

# Short Communication: Nanoscale heterogeneity of U and Pb in baddeleyite - implications for nanogeochronology and $^{238}$U series alpha recoil effects

Steven Denyszyn[1], Donald W. Davis[2], Denis Fougerouse[3]

[1]Earth Sciences, Memorial University of Newfoundland, St. John's, A1A 0G3, Canada
[2]Earth Sciences, University of Toronto, Toronto, M5S 3B1, Canada
[3]Geoscience atom probe facility, John de Laeter Centre and School of Earth and Planetary Sciences, Curtin University, Perth Australia

*Correspondence to*: Donald W. Davis (dond@es.utoronto.ca)

**Abstract.** Atom probe tomography (APT) of $^{238}$U and $^{206}$Pb has been applied to baddeleyite crystals from the Hart Dolerite (1791 ± 1 Ma) and the Great Dyke of Mauritania (2732 ± 2 Ma) in an effort to map U and Pb concentration at the nanometre scale. The purpose was to constrain the average nuclear recoil distance of $^{238}$U-series daughter nuclei in order to correct U-Pb ages determined on small baddeleyite crystals for alpha-recoil loss of Pb. Both crystals were thought to expose natural crystal

surfaces providing a boundary where maximum effects of recoil loss could be observed. The Hart Dolerite sample showed no variations in Pb concentrations near the edge. The Great Dyke sample shows U zoning and the associated $^{206}$Pb zoning is affected by alpha recoil, apparently adjacent to a natural grain surface. The sample also shows 10 nm-scale apparently primary clusters of U atoms that contain about 40% of the U. These are too small to constrain alpha recoil distance beyond a few nm but are apparently primary and their formation mechanism poses a dilemma. To constrain alpha recoil distance, a forward

modelling approach is presented where $^{206}$Pb redistribution functions were determined for a range of possible distances and synthetic $^{206}$Pb/$^{238}$U profiles were determined from the convolution of the observed U profile with the redistribution functions that were compared to the observed $^{206}$Pb/$^{238}$U profile. A complication is the fact that the 40 by 400 nm size of the sample is lower than the range of possible alpha recoil redistribution effects so it was necessary to extrapolate the observed U zoning. An oscillatory pattern gives the best fit to the observed profile but any reasonable extrapolation constrains the average alpha

recoil distance to be close to 80-90 nm, which is much larger than previous estimates using other methods. Either recoil distances can be highly anisotropic within small crystal samples or surface roughness was a factor that modified the recoiled Pb distribution. APT is a potentially useful approach to determining average alpha recoil distance but requires sampling of primary, smooth crystal faces.

# 1 Introduction

Atom probe tomography (APT) can be used to directly observe the distribution of individual atoms of elements in a mineral crystal lattice (Reddy et al., 2020). APT has been applied to baddeleyite in previous studies, mostly using the





baddeleyite mineral standard Phalaborwa (Reinhard et al. 2018) and/or crystals that underwent shock metamorphism (White et al. 2019; White et al. 2017). One germane finding of these studies is that while major elements are homogeneously distributed at the atomic level in baddeleyite, radiogenic Pb and incompatible trace elements (Si, Mg, Al, Yb and Fe) can be heterogeneously distributed in nanoscale domains (White et al 2017; White et al 2018). The nanoscale U distribution in Phalaborwa baddeleyite crystals was also shown to be variable with localized zonation, and heating baddeleyite to temperatures up to 500ºC does not seem to affect that distribution (White et al 2017).

Baddeleyite (monoclinic $ZrO_2$) is a common geochronometer in silica-undersaturated rocks that can be dated with <0.1% age precision using isotope dilution thermal ionization mass spectrometry (ID-TIMS). Like zircon (tetragonal $ZrSiO_4$), U is incorporated into its crystallizing lattice, but not Pb. Unlike zircon, radiation damage from the decay of U and its radioactive daughters has minimal effect in the form of Pb loss (Rioux et al., 2010; Lumpkin, 1999). However, minor discordance (<3%) between the $^{206}Pb/^{238}U$ and $^{207}Pb/^{235}U$ systems is common in baddeleyite of all ages, outside of uncertainties in U decay constants (Rioux et al., 2010; Schoene et al., 2006). Apparent daughter loss in younger samples (<500 Ma for ID-TIMS), where age interpretations must be based on $^{206}Pb/^{238}U$ ratios rather than $^{207}Pb/^{206}Pb$ ratios, is manifested as younger $^{206}Pb/^{238}U$ dates but data are still generally concordant due to relatively large $^{207}Pb/^{235}U$ uncertainties. This is because at young ages discordia lines become subparallel to the concordia curve.

Baddeleyite does not appear to be susceptible to low temperature alteration like radiation-damaged zircon, although it can break down to polycrystalline zircon with accompanying Pb loss during metamorphism (Davidson and van Breemen, 1988). This may be because its chemical stability is due to its composition whereas that of zircon is due to its structure, which is disrupted by radiation damage. Therefore, pre-treatment methods such as air abrasion (Krogh, 1982) or annealing and HF-leaching ("chemical abrasion"; Mattinson, 2005) that are designed to remove alteration and are frequently employed to mitigate Pb loss in zircon are generally unnecessary or ineffective (e.g., Rioux et al., 2010). Various mechanisms have been proposed for observed discordance in baddeleyite. They include micro- or cryptocrystalline zircon overgrowths, which are susceptible to extensive Pb loss (e.g., Darling et al., 2016; Schmieder et al., 2015); loss of intermediate daughter $^{222}Rn$ through diffusion (Heaman & LeCheminant, 2001); excess intermediate daughters $^{231}Pa$ (Ivanov et al., 2021; Sun et al., 2020; Amelin & Zaitsev, 2002) and $^{230}Th$ (Wu et al., 2015); and daughter loss through alpha recoil (Denyszyn et al., 2009; Davis & Sutcliffe, 1985). In particular, apparent preferential $^{206}Pb$ loss leading to a biased effect on both $^{206}Pb/^{238}U$ and $^{207}Pb/^{206}Pb$ ratios has been observed with improved precision on single analyses, and has been attributed to excess $^{231}Pa$ (Ibañez-Mejia and Tissot, 2019) and/or $^{222}Rn$ mobility (Pohlner et al., 2020). An observed relationship between grain size, specifically surface area-to-volume ratio, and degree of $^{206}Pb$ loss where crystal rims are more strongly affected than cores, implies either fast-pathway/volume diffusion of $^{222}Rn$ (Pohlner et al., 2020) or, most likely, alpha recoil as a cause.

Alpha recoil refers to the displacement of daughter radioisotopes as a result of the ejection of an alpha particle from the parent. The resulting lattice damage to the mineral's crystal structure facilitates Pb loss after low-temperature alteration, a phenomenon well-documented in zircon (e.g., Nasdala et al. 2010). Pb and other intermediate daughters in the U decay series can also be directly ejected from the crystal. Recoil distances for single alpha decays have been calculated to range from 20 to



33 nm in zircon (Nasdala et al. 2001), and while the directions of the alpha emission and recoil are random, an average cumulative recoil distance from the 8 decays of the $^{238}$U decay series should be about 3 times the average single recoil distance. There are 7 decays in the $^{235}$U decay series. Therefore, a zone of U-daughter depletion can be expected to make up the outer 70 ca. 50-100 nm of a given zircon crystal (Davis & Davis 2018), representing a volume in which alpha-recoil loss can be significant. Even in zircon crystals that are relatively equant compared to the flat, bladed habit of baddeleyite, and therefore have relatively low surface area-to-volume ratios, alpha recoil can generate measurable discordance (Romer 2003). For a crystal of 100 µm in longest dimension and 2:1 aspect ratio, this depletion zone can make up ca. 0.3% of the volume of intact crystals which have not undergone physical or chemical abrasion (Schmieder et al. 2015).

75 As alpha recoil is a fundamental physical process that must occur with every radioactive decay, determining average alpha-recoil distance is important to establish the degree to which discordance in baddeleyite can be attributed to this process. The alpha recoil distance has implications for the minimum useful volume of baddeleyite for geochronology, and for the selection of subsamples for spatially-detailed analysis (e.g., FIB-TIMS, White et al. 2020) because recoil will redistribute radiogenic Pb from internally zoned U. Accurate knowledge of the recoil distance would allow at least first order corrections 80 to ID-TIMS Pb/U ages based on the dimensions of samples.

Determining alpha recoil distances in baddeleyite from laboratory measurements is challenging. Denyszyn et al. (2009) tested the requisite correlation of grain size and apparent age of baddeleyites on the order of 10 µm thickness, and could not resolve any effect on $^{206}$Pb/$^{238}$U ratios at the scale of typical ID-TIMS U-Pb analyses, suggesting that recoil distances in baddeleyite are shorter than those for zircon. The strongest effects of alpha recoil should be seen at a natural baddeleyite grain 85 surface where there is a large reduction in U concentration from inside to outside the crystal. Pb concentrations within the crystal, hence $^{206}$Pb/$^{238}$U, should have dropped by a factor of 2 at the crystal surface (Davis and Davis, 2018). Thus far, the only direct measurement is by Davis & Davis (2018) who used a sensitive high-resolution ion microprobe (SHRIMP II) to create depth profiles of $^{206}$Pb/$^{238}$U ratios away from natural crystal surfaces in baddeleyite. Measurements were made on a total of 5 spots from 2 natural grain surfaces from the 2059 Ma Phalaborwa carbonatite (Heaman, 2009). They obtained a value of 90 24 ± 7 nm for the average recoil distance, which is similar to that calculated for zircon. The SHRIMP measurements were difficult because of the strong dependence of Pb/U biases on the condition of the sputtered surface in secondary ion mass spectrometry. As the hole deepens, the bias must be corrected by comparison with results from analysis of a polished crystal surface of a grain from the same sample. U concentrations were also found to drop off within grains near natural surfaces during some analyses, which was unexpected, but the average recoil distance was found to be about the same whether or not 95 this was taken as an artefact.

Alpha recoil results in a distribution of radiogenic Pb atoms that differs from the U distribution in having smaller concentration gradients. As mentioned, the largest U gradient should be encountered immediately at the crystal surface. The degree of broadening of the Pb distribution away from a natural surface can be calculated for an assumed average alpha recoil distance and compared to the measured profile, thus providing a way to potentially constrain its value as was done by Davis 100 and Davis (2018) but APT should be less affected by biases in U/Pb measurements with depth of analysis and can also be used



in the absence of a natural surface provided there is sufficient U variation due to zoning. We therefore applied APT to two baddeleyite crystals of known age and constructed profiles of U and radiogenic Pb distribution in order to constrain average alpha recoil distance in baddeleyite for the [238]U decay chain as well as study the distribution of U atoms.

## 2 Analytical and Modelling Methods

### 2.1 Sample preparation

Samples were selected based on identification of apparently well-defined crystal faces, concordant U-Pb analyses, and relatively old age and high U content to ensure measurable abundances of U and radiogenic Pb. Baddeleyite crystals were chosen from the Hart Dolerite (M2, [207]Pb/[206]Pb age of 1790.5 ± 1.4 Ma; Ramsay et al., 2019, their sample GS11043-1) and the Great Dyke of Mauritania (M5, upper-intercept age of 2733 ± 2 Ma; Tait et al., 2013, their sample GTD-8). All age errors are quoted at 95% confidence.

Both baddeleyite grains were picked and placed on individual scanning electron microscope (SEM) aluminum stubs covered by carbon tape (Fig. 1A and B). The stubs were sputter-coated with an approximately 200 nm thick layer of Cr, to serve both as a conductive layer and as a cap protecting the surface of the baddeleyite grains. Atom probe specimens were prepared from the Cr-coated grain with a Ga[+] Tescan Lyra3 Focused Ion Beam (FIB) at the Microscopy and Microanalysis Facility (MMF), Curtin University. A Pt layer was sublimated along the surface of the crystal in order to protect the region of interest from Ga implantation and Pt sublimation was used to fuse the needle-shaped specimen to the APT specimen holder. The FIB was operated at an accelerating voltage of 30 kV during the sculpting of the specimens and at 2 kV in the final stage to remove the external layer affected by the high-energy Ga beam. During the final stage of polishing, some Cr cap was intentionally left at the apex of the specimens to provide a conductive surface on the baddeleyite grain surface during FIB treatment, and to identify the original crystal surfaces (Fig. 2A and B).

### 2.2 Atom Probe Tomography (APT)

APT relies on the field evaporation of ions from a needle-shaped specimen. The original position of the ions in the analysed volume is given in three dimensions and at sub-nm resolution by a position-sensitive detector. The ions are identified by their time-of-flight and reported on a mass-to-charge ratio spectrum (mass spectrum in Dalton). This study used the Geoscience Atom Probe (Cameca LEAP 4000X HR), at Curtin University. The instrument was operated in laser assisted mode with a UV ($\lambda = 355$ nm) laser set at 100 pJ pulse energy and at a repetition rate of 125 kHz. The specimens were maintained at 50 K base temperature in order to inhibit surface diffusion during analysis. In the mass-to-charge ratio spectra, peaks higher than twice the background were identified and ranged for 3 dimensional reconstructions using Cameca's APsuite 6.3 software. Voltage evolution reconstructions were performed using a detector efficiency of 0.36, an image compression factor of 1.65 and a k-factor of 3.3. For baddeleyite, the atomic volume was calculated at 0.01133 nm$^3$/atom and the electric field was empirically determined at 29.08 V/nm (Fougerouse et al., 2022). Tomographic data for one specimen of each grain were successfully





acquired, with 62 million ions for specimen M5 (Great Dyke of Mauritania) and 65 million ions for specimen M2 (Hart Dolerite). Depth concentration profiles were generated for a 5 nm bin size and exclude the Cr cap.

The U and Pb isotopic compositions were quantified from the atom probe data by using a narrow range (0.1 Da) for each isotopic ionic specie. Uranium was spread over several ionic species with the dominant $^{238}UO_2^{++}$ (135 Da), but also $^{238}UO_2^{+}$ (270 Da) and $^{238}UO^{+++}$ (84.7 Da). No peaks were visible above background for $^{235}U$. Lead was present as $^{206}Pb^{++}$ and $^{207}Pb^{++}$ (103 and 103.5 Da, respectively). The local background signal was measured by selecting a wide 'peak free' region (0.5 to 1 Da) in proximity to each ionic specie and normalized to the width of the specie range (0.1 Da). Uncertainties were estimated

at 1σ using counting statistics and propagating background corrections adapting protocols previously defined for baddeleyite, zircon and monazite (Fougerouse et al. 2018; White et al. 2017; Peterman et al. 2016). A summary of APT results is presented in Supplementary Data File 1.

**2.3 Alpha Recoil Modelling**

Modelling of the $^{206}Pb$ concentration was carried out based on the observed $^{238}U$ distribution and assumed values of R, the average value of the recoil distance for each of the 8 alpha emitting nuclides. This was done by calculating an estimation of the redistribution function for a given R value and calculating the convolution of this function with the observed U distribution. As in Davis and Davis (2018), the assumed planar symmetry of the zoned U distribution simplifies the calculations, allowing the results to be expressed as 2 dimensional profiles where the redistribution function varies only with distance along the axis,

normal to the plane of zoning.

An average recoil distance is used for simplicity in formulating the problem but it does not describe the true kinetics of a recoiling nucleus. Each of the 8 nuclei in the $^{238}U$ chain decay with different energies and so will have different average recoil distances. Also, recoiled nuclei transfer energy to host atoms by multiple scattering events, which is the source of radiation damage, so their actual paths are complicated and cover a range of values even for the same decay. None of this is relevant to

the problem of correcting $^{206}Pb/^{238}U$ ages, which requires knowledge of the dimensions of the analyzed crystals and the $^{206}Pb$ redistribution curve, a function of the average total displacement of a $^{206}Pb$ daughter atom from its decayed $^{238}U$ nucleus. Thus, even if the R value is an average, it is useful for characterizing their overall displacement.

To approximate the redistribution function for a given alpha recoil distance, we start with a uniform plane of atoms and calculate the displacement of an atom normal to the plane after being displaced a distance R in a random direction. This is

done on a population of 1000000 atoms and the shape of the resultant curve of number of atoms versus normal distance, when normalized to an area of 1, approximates the redistribution function for the distance R (Fig. 3A). Calculations of redistributed Pb profiles were done using Visual Basic for Applications (VBA) in Excel. See Supplementary Data File 2 for examples. This contains software and instructions for repeating the calculations with different values of R.

The distribution functions after 8 recoils approximate a Gaussian curve (Fig. 3B), which may seem surprising since, as

shown in Suppl. Data File 2, the result of 1 recoil is a parabolic-like function and even 4 recoils produce a function that is





noticeably different from a Gaussian. Convergence to a Gaussian shape by recursion of any random process, even one whose distribution is itself non-Gaussian, is predicted by the well-known Central Limit Theorem (Bárány and Vu, 2007).

Alpha recoil results in spreading the decayed U-series atoms within a given distance bin into adjacent bins, which involves calculating a convolution of the observed U profile with the above determined area-normalized redistribution function. The

result gives a theoretical recoiled $^{206}$Pb distribution and profile of $^{206}$Pb/$^{238}$U ratios with the assumed value of R. Note from Fig. 3 that the mean total displacement of a Pb atom is much greater than the R value because there are 8 decays. Fig. 4 and Suppl. Data File 2 illustrate the result of eight alpha recoils with an R of 40 nm for a hypothetical Gaussian-shaped U concentration of similar scale to the peak from the sample. For simplicity, it is assumed that the amount of $^{238}$U at present is equal to the amount of $^{206}$Pb produced. Convolution of a Gaussian curve produces a wider Gaussian curve. The $^{206}$Pb/$^{238}$U

profile shows roughly symmetrical peaks on each side of the U peak where excess $^{206}$Pb has been projected from adjacent high U levels on each side of the peak onto part of the tail, raising the ratio from 1, while within the U peak the ratios are less than 1.

## 3 Results

### 3.1 U and $^{206}$Pb/$^{238}$U zoning from APT

The U concentration in specimen M5 (Mauritania) is higher than in specimen M2 (Hart) with an average U of 288 ppma (parts per million atomic) for M5 and 23 ppma for M2. In specimen M5, the distribution of U shows a gradient of concentration from low U content near the surface and higher content towards the centre of the grain (Fig. 5), varying from approximately 150 ppma to 650 ppma.

The Cr cap coating preserved at the tips of the atom probe specimens confirms the apparent position of the crystal surfaces (Fig. 2). Surprisingly, profiles of background-corrected abundances of Pb and U for each crystal (Fig. 5) do not show the expected sudden drop-off of Pb in the outermost 10-50 nm as seen in previous SHRIMP measurements (Davis and Davis, 2018). For specimen M5, the abundances of these elements is non-uniform, with a zone of relative enrichment in the 200-400 nm depth range, which is interpreted to represent growth zoning of U (Figs 5 and 6A). The sampled surfaces of the baddeleyite

crystals appeared to be natural crystal surfaces based on visual inspection under SEM imaging and the absence of obvious fracturing. In the Davis and Davis (2018) study, one out of the three studied grains also did not show decreasing Pb near the crystal boundary and it was concluded that the boundary may have been a cleavage plane, which may also be the case for sample M2.

### 3.2 Alpha recoil modelling results from U zoning

Sample M2 shows a uniform U concentration (Fig. 5) and is of no use for constraining alpha recoil distance beyond the conclusion that if it were sampled at a crystal boundary the recoil distance must be very short. If not, the step in U concentration along the axis of sample M5 (Fig. 6A) should result in Pb concentration effects dependent on alpha recoil distance. M5 also



has more radiogenic Pb because it is the oldest and has the higher U concentration, which makes parent/daughter ratios easier
to measure.

The measured U concentration and $^{206}Pb/^{238}U$ ratio profiles in M5 are shown in Fig 6B. The measured age of M5 corresponds to an equilibrium radiogenic $^{206}Pb/^{238}U$ ratio of 0.53. The measured $^{206}Pb/^{238}U$ profile shown in Fig. 6B is roughly constant and close to the equilibrium value for distances of about 1 nm to 150 nm, which corresponds to the low U zone. It then drops to a fairly constant value notably lower than equilibrium over the peak of the high U zone. If the recoil distance
were very small (less than a few nanometres), Pb would not be significantly displaced from U and one would expect the $^{206}Pb/^{238}U$ profile to be constant, independent of the U concentration, which is not the case. If the recoil distance were very large, Pb would be redistributed approximately independent of the U profile and should have a constant composition. In this case, the ratio profile would be proportional to the inverse of the U concentration. The measured profile does show that the $^{206}Pb/^{238}U$ ratios are low where the U concentration is high and high where it is low but the U concentrations differ by a factor
of over 3 whereas the ratio levels differ by about 1.5. Therefore, alpha recoil must have significantly modified the daughter Pb distribution.

The measured U profile is not wide enough to avoid effects of recoil beyond its measured range. It is therefore necessary to extrapolate the profile above and below. Given that there are two unknowns, one of which is an array, the determination of R will not be unique but it may be possible to constrain it by reasonable assumptions and examining how different U
distributions affect the shape of the ratio profile.

The shape of the measured $^{206}Pb/^{238}U$ profile constrains likely adjacent U concentrations as well as alpha recoil distance. It is remarkable that the $^{206}Pb/^{238}U$ ratio over the low U part of the zone is fairly constant and close to the equilibrium ratio. Alpha recoil should scatter excess Pb into this region from the high U part of the zone to the right which, by itself, should result in the higher than equilibrium ratio. The fact that the ratio is near-equilibrium requires that the U concentrations to the left (at
negative distances) be close to zero since any U in this region would add more Pb, tending to raise the measured $^{206}Pb/^{238}U$ ratio further. This strongly suggests that the U concentration at negative distance is zero and that the crystal face was a natural grain surface as assumed. Therefore, we assign a value of zero for U concentrations at negative distance with fair confidence. Another qualitative deduction is that the average alpha recoil distance must be quite large (>40 nm) to explain the relatively flat $^{206}Pb/^{238}U$ profile over the U peak and that it is so far below the equilibrium ratio.

Assuming that the U concentrations at <0 nm are zero and those >475 nm are projected linearly downward to the value at 0 nm, which is then kept constant (Fig. 7A), results in the models shown in Fig. 7B for different assumed values of average recoil distance. The distance that gives the best fit is about 80 nm. This fits the region over the U peak quite well and also matches the profile at the start of the low U section but deviates from the measured profile near the start of the high U peak because of accumulation of recoiled Pb. Flattening this section would require a larger recoil distance to smooth the distribution of Pb recoiled from the left, but this would cause more Pb to recoil out of the peak region, dropping its modelled $^{206}Pb/^{238}U$
ratio below the measured value. This can be offset by assuming a high-U region to the right of the measured profile. The simplest assumption is that of oscillatory zoning, as shown in Fig. 8. This seems to give the lowest values for mean square of





weighted deviations (MSWD) as shown in Fig. 9 where recoil distances between 80 and 90 nm give MSWD values of 1.3 to 1.2.  The modelled distributions straddle the observed scatter over the observed U peak but tend to be above the measured ratio

distribution over the low U part. The MSWD shows a local drop at higher recoil values around 120 nm probably because the profile flattens over the low U section and provides a better fit.   Modelled ratios for 160 nm fall below the measured profile over the U peak because of greater Pb loss due to recoil. Pb can be added by making the extrapolated cycle higher in U but this will increase the curvature of the profile unless recoil distance is made still larger. The results of modelling on these and a number of other extrapolated profiles are shown in Supplementary Data File 2. We were not able to match the measured

profile with MSWD = 1. In order to attain MSWD < 1.5 the average recoil distance must be assumed to be 80 nm – 90nm. It would be unrealistic for us to assign an error to the above estimate. Such an error might be assigned on the basis of the measurement errors in the data but it would be dependent on only one extrapolation model. The problem is that we cannot evaluate all possible models for extrapolation of the U distribution beyond the range of measurement but within the range of alpha recoil effects. It might be possible after a great deal of computation and the application of advanced algorithms, but this

would be beyond the scope of the work as well as unnecessary, since this estimate is unlikely to be accurate for reasons discussed below.

### 3.3 U clustering

About 40% of the U in specimen M5 occurs in clusters of ~10 nm diameter spaced approximately 20 nm apart.  The clusters

were quantified and averaged (Figure 10) using the proximity histogram method of Hellman et al. (2000). Results are shown in Figs 10 and 11. "Clusters" is used to describe the regions of relatively high concentration of the atoms of interest; "matrix" refers to the parts of the crystal between the clusters. On average, clusters are enriched in U (2,406 ppma), Pb (305 ppma), Ti (1,614 ppma) and Fe (2,400 ppma) compared to the matrix (U = 225 ppma; Pb = 145 ppma; Ti = 712 ppm; Fe = 2,086 ppma). Although enriched within the U clusters, the Pb distribution is more diffuse than U and the clumping less evident. The Hf

composition is unchanged between the clustered or matrix domains. The U clusters are observed in all parts of the dataset, even at U concentrations as low as 150 ppma (Fig. 10 and MP4 files in Supplementary Data showing rotating animations of the needle volumes with U and Pb atoms represented in 3D space). No clusters were observed in specimen M2, in which U and Pb were homogeneously distributed.

For specimen M5, the entire specimen yields a $^{206}Pb/^{238}U$ ratio of 0.528 ± 0.011 for a calculated age of 2,734 ± 70 Ma and

a $^{207}Pb/^{206}Pb$ ratio of 0.179 ± 0.017 for an age of 2,640 ± 162 Ma. The matrix domain of the sample yields a $^{206}Pb/^{238}U$ ratio of 0.772 ± 0.020 for a calculated age of 3,687 ± 128 Ma and a $^{207}Pb/^{206}Pb$ ratio of 0.165 ± 0.019 for an age of 2,507 ± 193 Ma. The combined clusters composition is 0.168 ± 0.007 for the $^{206}Pb/^{238}U$ ratio and a $^{206}Pb/^{238}U$ age of 1,002 ± 46 Ma and a $^{207}Pb/^{206}Pb$ ratio of 0.232 ± 0.042 for a $^{207}Pb/^{206}Pb$ age of 3067 ± 292 Ma. In this dataset, the $^{207}Pb^{++}$ peak suffered a high background due to the thermal tail of the $^{206}Pb^{++}$ peak, detrimental for the precise quantification of $^{207}Pb$. The Pb content of

specimen M2 was not high enough to calculate meaningful $^{206}Pb/^{238}U$ and $^{207}Pb/^{206}Pb$ ratios.



The distribution of Pb should reflect the effect of alpha recoil from the clustered U, generating more diffuse clusters of Pb around a U cluster. Convolution is unnecessary as the excess U on the cluster is effectively a point source with the only relevant parameter for recoiled Pb atoms being the distance from the cluster. However, the relative volumes of spherical shells around the cluster increase as the cube of their radius, which means that recoiled $^{206}$Pb atoms are rapidly diluted by Pb in the matrix
as their recoil distance becomes larger.

The averaged measured $^{238}$U and $^{206}$Pb concentrations and the $^{206}$Pb/$^{238}$U ratios within and outside the clusters are shown on Fig. 11A and 11B, respectively. Distance zero represents the location of the cluster boundary and distance is negative within the boundary. The volume and number of atoms are rapidly reduced as the centre of the cluster is approached, so measurement errors become large (Fig. 11A). Modelling alpha recoil from a cluster is shown in Supplementary Data File 3. The average
cluster is considered to have a radius of 3.5 nm, the radial bins are taken as 0.5 nm wide and the U and Pb concentrations for the first 3 bins are taken as being constant and set at the measured values for the third bin (Fig. 11B). Any measurements deeper than this are too imprecise to be meaningful.

Although there is an increase in $^{206}$Pb concentration within the cluster this is well below the amount that would give a $^{206}$Pb/$^{238}$U equilibrium ratio corresponding to the age of the sample (about 0.5). As mentioned above, the $^{206}$Pb/$^{238}$U ratio in the
interior of the cluster is about 0.17. This might be explained if the clusters formed at about 1 Ga and the recoil distance were very small (<1 nm) but there is no reason to expect U mobility at low temperatures, since it is considered to have a blocking temperature even higher than Pb, at least in zircon (Cherniak and Watson, 2003), or that the recoil distance is negligible (see above). There is also a narrow peak in the radial $^{206}$Pb/$^{238}$U ratio at about 5 nm, where it rises above 1 due to the fact that the U concentration decreases near the average cluster boundary. Away from the average cluster the background U and Pb
concentrations should reflect the equilibrium $^{206}$Pb/$^{238}$U ratio for the 2.7 Ga age of the sample but in fact the density of clusters is high enough to give the matrix a significantly higher value (see above).

## 4. Discussion

### 4.1 The meaning of alpha recoil constraints from APT

The 80-90 nm alpha recoil constraint is significantly higher than the average recoil distance of 24 ±7 nm found from depth profiling of natural surfaces of baddeleyite crystals using SHRIMP (Davis and Davis, 2018) as well as theoretical estimates for zircon (Nasdala et al. 2001). Thermal diffusion of Pb would have a similar effect to alpha recoil and would give a too-large recoil distance if it had occurred but this is not possible over the geologic history of the sample since the metamorphic grade does not exceed greenschist facies and post-crystallization heating occurred early in the history of the sample. The only other
possibility is if the sample were heated enough during FIB milling to allow significant Pb diffusion but this would likely require temperatures in excess of 1000°C. Effects of heating by FIB milling have not been seen in other minerals with lower melting points so the temperature must have been well below that required for diffusion of Pb during the ca. 2 minute period of milling.



If recoil distances in an ordered lattice are highly anisotropic this might result in a higher than average value normal to a
crystal face. This seems inconsistent with the result from SHRIMP data but the SHRIMP measurements were carried out with
a beam size of about 30 μm, whereas the APT length X-Y scale is about 3 orders of magnitude smaller, which equates to a
sample volume about 6 orders of magnitude smaller and therefore more likely to preserve a perfect lattice arrangement (Smith
and Newkirk, 1975) at a scale that is affected by alpha recoil. Another possible explanation is that surface roughness at the
scale of alpha recoil effects resulted in violation of the assumption that U distribution was uniform in the X-Y plane outside
of the sampled volume. Fig. 1B shows that the surface from which the sample was milled is not smooth at the scale of electron
microscopy and the wavelength of topographical features could be less than 1 micron. There is also a suggestion in Fig 3 that
the high to low U transition is not planar. It will be important in future APT studies to select samples from natural, preferably
(100), surfaces that appear smooth at the scale of alpha recoil effects (Kozikowski, 2020) and/or to assure that any U zoning
is uniform beyond the scale of alpha recoil effects.


### 4.2 U clusters

The small size of the clusters makes the Pb/U ratio profile very sensitive to recoils. The modelling shows that average recoil
distances above 1 nm effectively remove all radiogenic Pb from the cluster and dilute it into the background, although R
distances up to a few nm tend to pile recoiled Pb into the U trough, increasing the peak in the $^{206}$Pb/$^{238}$U profile slightly
(Supplementary Data File 4). Therefore the cluster distributions can only be used to constrain the average recoil distance as
being greater than a few nanometres. The presence of the small $^{206}$Pb peak within the average cluster, as well as the trough in
U concentration around it cannot be explained by alpha recoil. They are most likely to be primary effects of cluster formation
and the depletion of the silicate melt in proximity to the preferential sorption of the U ions in the clusters. The $^{206}$Pb peak might
correspond to elevated concentrations of common Pb, which might be confirmed with $^{204}$Pb measurements if this mass could
be effectively resolved. The trough in U concentration is perhaps due to freezing in of a U concentration gradient in the crystal
or melt following growth of the high-U clusters.

The U composition of the clusters in specimen M5 differs by approximately 0.25 at.% from the composition of the matrix.
Although the clusters are primarily composed of (Zr & Hf)O$_2$, they show elevated concentrations of Fe and Ti as well as U.
Thus, they are nano-scale baddeleyite domains enriched in trace elements. To the authors' knowledge, this is the first
observation of U clustering in a mineral used for geochronology. Fougerouse et al. (2016) showed apparent primary clusters
of Au atoms in arsenopyrite but these were explained as an aberration caused by evaporation from conductive gold nano-
inclusions within the crystal. Valley et al. (2014) measured clusters consisting of Pb, Y, and Yb (but not U) in a 4.4 Ga zircon
core that was overgrown at 3.4 Ga. In this case, they concluded that clustering was due to migration of some trace elements
through damaged crystal domains during metamorphism. Several mechanisms have been proposed for the formation of clusters
in other minerals by secondary processes including: annealing of radiation damage (Peterman et al., 2021; Verberne et al.,
2020); phase exsolution during cooling (Fougerouse et al., 2018); deformation (White et al., 2018; Fougerouse et al., 2019)
and fluid alteration (Joseph et al., 2023). The M5 sample was not subjected to metamorphism above greenschist facies (Ramsay




et al., 2019; Tait et al., 2013) nor is there evidence of fluid alteration, deformation or exsolution. Thus, the U distribution in M5 appears to be a result of primary crystallization and may be the first observation of primary clustering of trace elements in

any mineral.

The mechanism by which the clusters formed is unknown but oscillatory zoning of U, which also appears to be present in sample M5, may be a related process. There is insufficient information from the present experiment, such as observations of zoning patterns and extent of U clustering in different zones, to undertake a detailed analysis of either and it is difficult to envision any magmatic crystallization process that could result in nano-scale spherical clusters of trace elements. Any

secondary process would have to result in very early diffusion of U, which should require pervasive structural adjustment of the crystal. One possibility might be if the original crystal formed as ziroite, a recently identified tetragonal form of $ZrO_2$ (Ma et al. 2023). This has recently been found as micron-scale clusters associated with baddeleyite in melt inclusions in mantle-derived corundum. Its presence in a mantle-derived assemblage suggests that the tetragonal form of $ZrO_2$ is stable at high pressure. If the original crystal had formed at high pressure within the plume magma and been transported into the crust, the

baddeleyite crystals might spontaneously revert to their stable monoclinic low pressure form at magmatic temperature. Such a recrystallization might allow diffusion of U but it is unclear why U atoms would tend to coalesce.

The observation of surface roughness in Fig 2B across the 001 face suggests either that growth of this face was a highly irregular process, in contrast to the 100 and 010 faces, which appear smooth, or it is a broken surface. As mentioned above, the $^{206}Pb/^{238}U$ profile along with alpha recoil modelling suggests that U concentration outside the surface was very low,

although this is not definitive proof that it is not a broken surface. Clearly more grains from this sample need to be examined along with APT analysis of U distributions next to crystal surfaces and within crystals.

## 5 Conclusions

The original aim of the experiment was to determine the Pb depletion profiles near natural faces of two baddeleyite crystals from separate samples. In the case of the Hart dolerite baddeleyite (M2) no such profile was observed, suggesting that the boundary was cleavage related. In the case of baddeleyite from the Great Dyke of Mauritania (M5), the measured profile strongly suggests that it was a natural crystal boundary, but the zoned distribution of U obscures the normal drop-off in Pb/U that would be expected. We present a different method of using atom probe tomography of baddeleyite in order to constrain

the average recoil distance from U decay. Preliminary results suggest an R of about 80-90 nm, based on the assumption that the observed U compositional gradient was due to internal zoning, is uniform on the X-Y plane, and the modified radiogenic Pb compositional gradient is due to alpha recoil. A more precise estimation is prevented by the limited size of the APT sample, which prevents knowledge of the composition over the entire range affected by recoil. The >80 nm estimate is much higher than that obtained from depth profiling of natural baddeleyite crystal surfaces using SHRIMP (24 ± 7 nm, Davis and Davis,

2018). This might be explained if recoil distances are strongly anisotropic and a high degree of order was preserved adjacent to the crystal boundary at the small scale of the sample. Alternatively, surface roughness may have modified the recoil





distribution, resulting in an anomalously high apparent recoil value. APT is an effective tool that could provide a much better constraint from analysis of well-characterized crystal boundaries with a more uniform U concentration profiles but it is important to establish that the sampled grain surface is smooth at the scale of alpha recoil effects.

This sample also shows evidence of primary U clusters with radii of a few nanometres. We conclude that these clusters are primary, formed during of close to initial crystallization of the baddeleyite. Depletion of radiogenic Pb within the clusters constrains alpha recoil distances to be more than a few nanometres. The clusters nevertheless show anomalously high Pb, although much less than for radioactive equilibrium, as well as high Fe and Ti concentrations, and U depletion around their margins. Explaining their formation remains a challenge for crystal chemistry. Though it is impossible to extract meaningful

U-Pb ages from limited (ca. 100 nm) regions of the crystal due to Pb redistribution, when using the entire APT volume meaningful ages can be calculated that correspond to those obtained using the ID-TIMS method.

**Acknowledgements.**

Ulf Söderlund (Lund University) is thanked for supplying samples of baddeleyite from the Great Dyke of Mauritania. We
thank Alyssa McKanna, Alberto Pérez-Huerta and Michelle Foley for constructive reviews that improved the manuscript.

**Code and data availability**. The code and supplementary data repository are found at:

https://osf.io/yp6n7/?view_only=1ac1a8cb60cf42028b146653b0c7e7e2


**Author contributions.**

SD conceived the project, furnished the samples, contributed to analysis. DF acquired and processed Atom Probe data. DWD wrote software to simulate recoil models and compare to data. All authors contributed to writing the manuscript.

**Competing interests.**

There are no competing interests.

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





**Figure Captions**

Figure 1: Scanning electron photomicrographs of the two studied baddeleyite crystals: A = Hart Dolerite, B = Great Dyke of Mauritania. Rectangles denote apparent fresh crystal faces; higher-relief feature is the platinum strip applied to protect the selected region from gallium implantation and to facilitate handling of the selected area after being carved out by the focussed ion beam.

Figure 2: Scanning electron micrograph photomicrographs of atom probe tomography specimens from (A) sample M5 of the

Great Dyke of Mauritania and (B) sample M2 of the Hart Dolerite. The observed crystal surfaces are beneath the Cr cap.

Figure 3: Distribution of $^{238}$U and $^{206}$Pb atoms in samples M5 and M2 from atom probe tomography. The observed crystal surfaces are at the upper parts of the distributions (<0 nm distance).

Figure 4: (A) measured $^{238}$U % concentration along the Z axis of sample M5. (B) Measured $^{206}$Pb/$^{238}$U ratio along the Z axis of sample M5. Dotted line represents the equilibrium $^{206}$Pb/$^{238}$U value for the age of the sample.

Figure 5: (A) Redistribution curves determined for 8 alpha recoils of 1,000,000 atoms with assumed recoil distances (R) of 20, 30 and 40 nm. The curves are normalized to an area of 1. (B) Comparison of redistribution curve for R = 40 nm with a Gaussian function with sigma of 82 nm. The high-U and -Pb zone labelled "Cap" indicates the cap of Cr coating applied to the crystal before FIB milling.

Figure 6: (A) Hypothetical Gaussian shaped $^{238}$U distribution peak on a constant background and resulting $^{206}$Pb distribution

after 8 alpha recoils with assumed recoil distance of 40 nm for each. (B) Resulting $^{206}$Pb/$^{238}$U ratio profile.

Figure 7: (A) Measured and linearly extrapolated U distribution. (B) Measured versus modelled $^{206}$Pb/$^{238}$U profiles for different assumed values of average recoil distance across the measured distance range, assuming the full U concentration profile shown in A.

Figure 8: (A) Measured and extrapolated U distribution. (B) Measured versus modelled $^{206}$Pb/$^{238}$U profiles for an assumed

value of average recoil distance of 40 nm with the single high-U zone shown in A.

Figure 9: Mean squares of weighted deviations (MSWD) for modelled versus measured $^{206}$Pb/$^{238}$U profiles as a function of the assumed average recoil distance, assuming the oscillatory zoned U distribution shown in Fig 8A.

Figure 10: (A) Distribution of 0.2 at% U isoconcentration surfaces delineating U clusters in sample M5. The high-U region at the tip of the needle indicates the cap of Cr coating applied to the crystal before FIB milling. (B) Distribution of U atoms

within a small region.

Figure 11: (A) Average $^{238}$U and $^{206}$Pb profiles around U clusters in sample M5. The two U and Pb values at the highest positive distances (inside of cluster) are too imprecise to be meaningful. (B) Measured average $^{206}$Pb/$^{238}$U profile around U clusters in sample M5. The blue line marks the equilibrium $^{206}$Pb/$^{238}$U ratio. Modelling with assumed average recoil (R) values above 1 nm produces similar profiles so clusters cannot be used to constrain realistic values of R.

Figure 13: $^{206}$Pb recoil profile from a uniform U distribution at a large baddeleyite crystal face (001) from a crystal 10 microns wide with assumed average recoil distances R of 0 nm (no redistribution) and 80 nm.





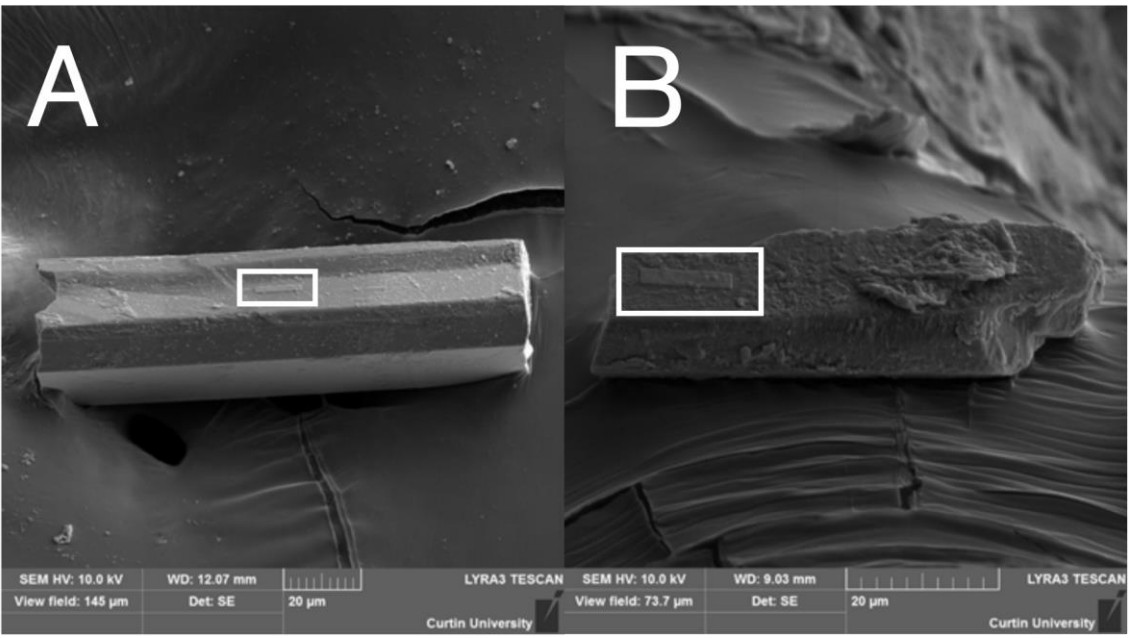

**Figure 1: Scanning electron photomicrographs of the two studied baddeleyite crystals: A = Hart Dolerite, B = Great Dyke of Mauritania. Rectangles denote apparent fresh crystal faces; higher-relief feature is the platinum strip applied to protect the selected region from gallium implantation and to facilitate handling of the selected area after being carved out by the focussed ion beam.**

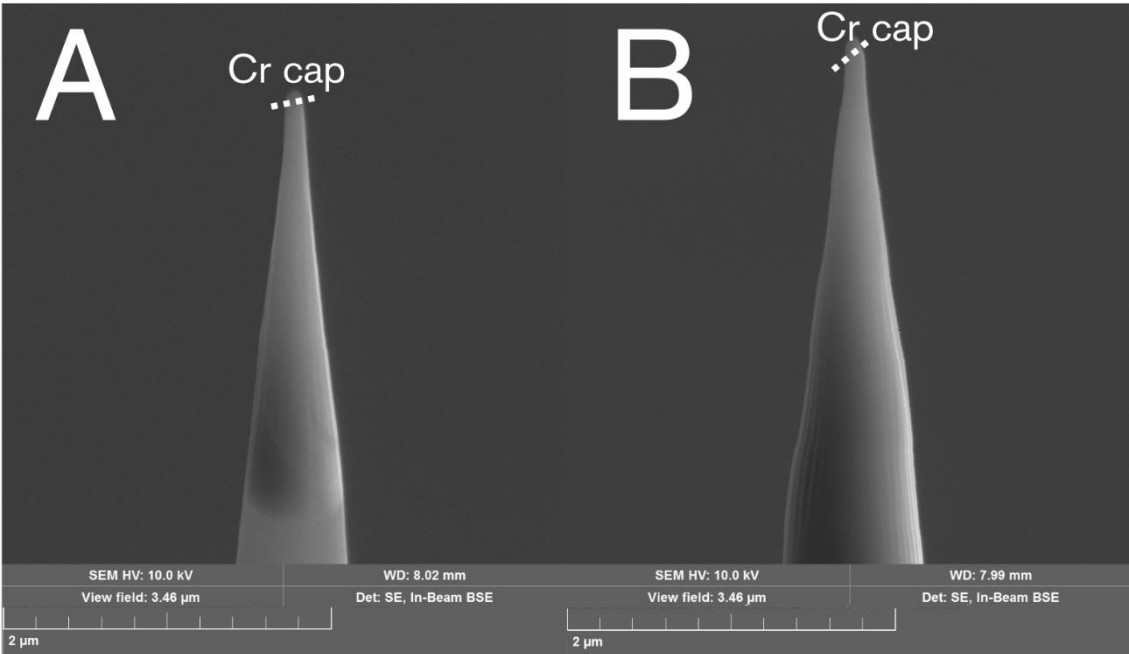

**Figure 2: Scanning electron micrograph photomicrographs of atom probe tomography specimens from (A) sample M5 of the Great Dyke of Mauritania and (B) sample M2 of the Hart Dolerite. The observed crystal surfaces are beneath the Cr cap.**







Figure 3: (A) Redistribution curves determined for 8 alpha recoils of 1,000,000 atoms with assumed recoil distances (R) of 20, 30 and 40 nm. The curves are normalized to an area of 1. (B) Comparison of redistribution curve for R = 40 nm with a Gaussian function with sigma of 82 nm.






**Figure 4: (A) Hypothetical Gaussian shaped** $^{238}$**U distribution peak on a constant background and resulting** $^{206}$**Pb distribution after 8 alpha recoils with assumed recoil distance of 40 nm for each. (B) Resulting** $^{206}$**Pb/**$^{238}$**U ratio profile.**




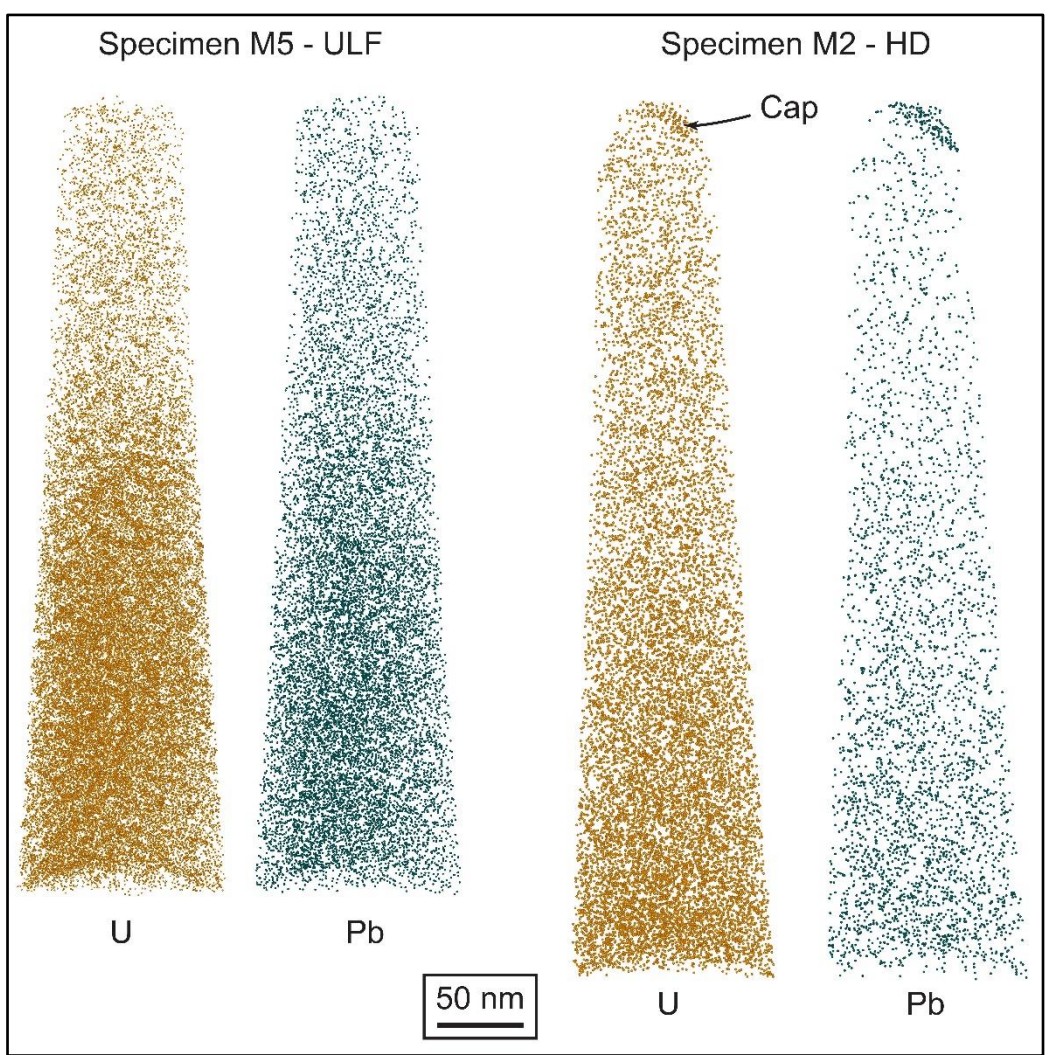

**Figure 5: Distribution of $^{238}$U and $^{206}$Pb atoms in samples M5 and M2 from atom probe tomography. The observed crystal surfaces are at the upper parts of the distributions (<0 nm distance). The high-U and -Pb zone labelled "Cap" indicates the cap of Cr coating applied to the crystal before FIB milling.**







**Figure 6: (A) measured** $^{238}$**U % concentration along the Z axis of sample M5. (B) Measured** $^{206}$**Pb/**$^{238}$**U ratio along the Z axis of sample**
**M5. Dotted line represents the equilibrium** $^{206}$**Pb/**$^{238}$**U value for the age of the sample.**







**Figure 7: (A)** Measured and linearly extrapolated U distribution. **(B)** Measured versus modelled $^{206}Pb/^{238}U$ profiles for different assumed values of average recoil distance across the measured distance range, assuming the full U concentration profile shown in A.







**Figure 8: (A)** Measured and extrapolated U distribution. **(B)** Measured versus modelled $^{206}Pb/^{238}U$ profiles for an assumed value of average recoil distance of 40 nm with the single high-U zone shown in A.

 

**Figure 9: Mean squares of weighted deviations (MSWD) for modelled versus measured** $^{206}$**Pb/**$^{238}$**U profiles as a function of the assumed average recoil distance, assuming the oscillatory zoned U distribution shown in Fig 8A.**





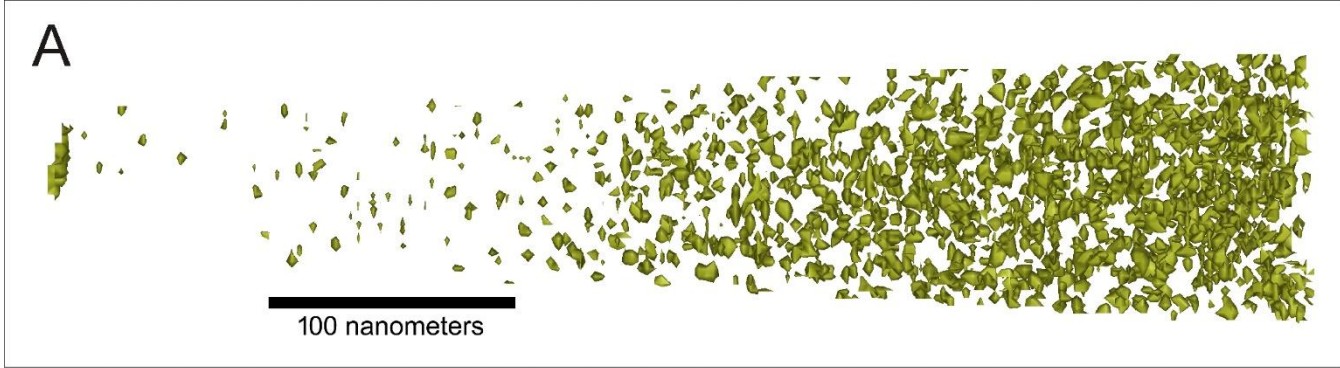

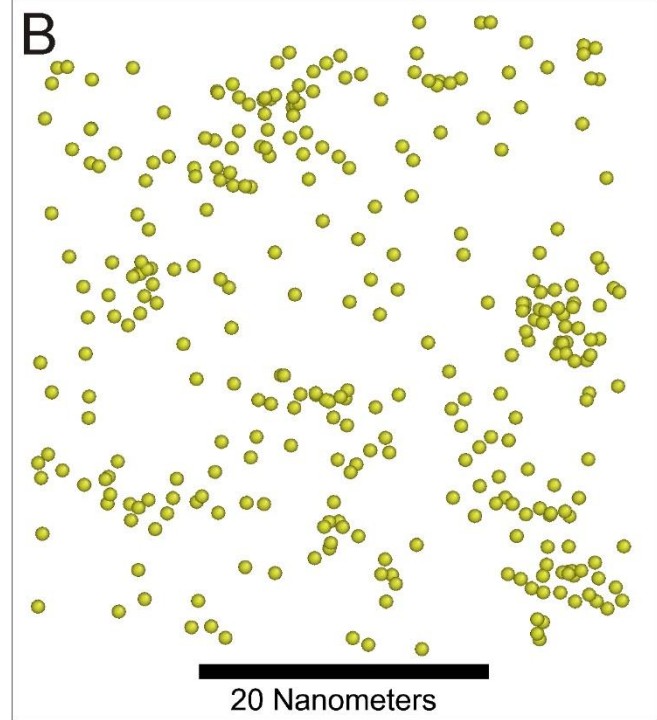


**Figure 10: (A) Distribution of 0.2at% U isoconcentration surfaces (about ten times background) delineating U clusters in sample M5. The high-U region at the tip of the needle indicates the cap of Cr coating applied to the crystal before FIB milling. (B) Distribution of U atoms within a small region.**






**Figure 11: (A)** Average $^{238}$U and $^{206}$Pb profiles around U clusters in sample M5. The two U and Pb values at the highest positive
distances (inside of cluster) are too imprecise to be meaningful. **(B)** Measured average $^{206}$Pb/$^{238}$U profile around U clusters in
sample M5. The blue line marks the equilibrium $^{206}$Pb/$^{238}$U ratio. Modelling with assumed average recoil (R) values above 1 nm
produces similar profiles so clusters cannot be used to constrain realistic values of R.