# Peer review of "Short Communication: Nanoscale heterogeneity of U and Pb in baddeleyite - implications for nanogeochronology and 238U series alpha recoil effects"

_Geochronology, 2024_

## Referee Comment (RC1)

**Overview**

Denyszyn and others couple atom probe tomography (APT) and numerical modeling to examine $\alpha$-recoil processes in the U-Pb system of baddeleyite. They find that only one of the two samples (Great Dyke of Mauritania) shows any heterogeneity in U and Pb distribution, and they disregard the other sample (Hart dolerite), concluding that it is an interior region of a crystal exposed along a cleavage plane. Through APT, the authors identify both $^{238}$U and $^{206}$Pb profiles that reflect a combination of crystallization and recoil effects. They estimate a mean recoil length for the $^{238}$U-series (80–90 nm) that is larger than prior estimates and identify some plausible causes for this discrepancy.

Overall, I find the work and conclusions very sound, and I find this to be an important and relevant contribution to the U-Pb and U-series communities' understanding of recoil processes at a very fine scale [1]. My primary criticism of this work is the lack of clarity and organization in the figures, captions, and some of the text, which collectively make it difficult to efficiently interpret the authors' findings.

Because the scientific work is sound and the conclusions relevant to the field of geochronology, I would recommend the manuscript for publication in *Geochronology* if the following comments are sufficiently addressed. In addition to these specific comments, I encourage the authors to review the manuscript with special attention to clear figures, prose, and descriptive figure captions.

Graham H. Edwards

**General Comments:**

- The introduction is well-written and provides a very thorough background.

- I found it cumbersome to interpret figures with unlabelled y-axes. While the labels were floated in the plots as text boxes, the authors should label axes directly wherever possible (e.g. Figs. 3, 4, 6–9, 11) and position the y-axis and corresponding tick labels outside of the plot area to ensure they are as legible as possible (Figs. 3,4,7). I recognize that the authors prepared their figures in Excel, which offers limited customization capabilities, but all of these edits are possible in Excel and will significantly improve figure readability.

- The authors refer with some frequency to Supplementary Data, including three spreadsheets. While I take no issue with referring to this supplementary data, these spreadsheets are poorly curated and have inadequate metadata. In the case of Supplementary Data File 3, this does not appear to contain the data the authors describe on line 274.
* * *
[1]My expertise is in the realm of U-series recoil processes and U-Pb geochronology. I have limited experience in baddeleyite crystallization processes and APT. My limited commentary on those topics reflects my naïveté, and I trust that referees more expert in these topics than myself can provide constructive commentary.

Moreover, Supplementary Data Files 2 and 4 are multi-page xlsm files. For clarity, the key plots should be presented as figures with descriptive captions so that readers can efficiently interpret the authors' key points. I think it is good practice to include the xlsm files to illustrate the methodological process, but these should not be the primary format of presenting/describing nuanced supplementary data.

- For clarity in the results and discussion, I advise the authors to use a consistent and specific set of directions (e.g. edge, interior) that consistently describe orientation relative to the original sample rather than terms like "left" (lines 219–30) that are dependent on the orientation of the APT specimen.

- Some of the text-based figure captions (lines 510–41) do not correspond with the current figure numbers. In general, the authors should double check that captions and in-text references correspond with appropriate figures and supplementary data files.

- I find the use of a convolution of the U distribution with the redistribution distribution appropriate for estimating the distribution of recoil transported radiogenic Pb. However, I think there should be a more extended discussion at the beginning of §2.3 justifying this approach. Lines 168–9 give this a good start, but I think most readers would benefit from more detail on this specific method.

- The Cr caps appear to be at angles to the z-axis of the FIB-milled specimens. This is curious, as the authors model systems with the z-axis orthogonal to the crystal surface. The authors should comment on how the needle-shaped specimens are oriented relative to the surfaces of grains they were milled from. Are the caps just apparently skewed or does this reflect the angle of the needles relative to the crystal surface.

- The authors conclude that the Hart dolerite is an exposed cleavage plane and do not consider it further. However, they consider spaces between bladed crystals as fast diffusion pathways of atom loss by recoil. They should comment on how these specific systems differ. (Presumably, the cleavage plane was exposed by a very recent breakage and was within strong crystal lattice previously, but I still think an explicit statement is worthwhile).

**Comments by line #:**

**97** I think this is based on the observations of Davis and Davis (2018), right? I think it's worth referencing the relevant study again with this statement.

**201–10** Please mention how the abundances across the transect are calculated. Are they calculated from the entire disk of each depth bin? Also please mention that a distance of zero refers to the tip of the specimen (and the corresponding crystal surface). I could figure it out by comparing figs. 5 and 6A, but an explicit statement in the text and corresponding figure captions (e.g. 6) would be very helpful.

**228–9** Please elaborate on the statement "but deviates from the measured profile near the start of the high U peak because of accumulation of recoiled Pb."

**230** Wouldn't $^{206}$Pb recoiled from the left be relatively inconsequential compared to Pb recoiled from the $^{238}$U peak? The process should be described in more detail.

**274** Supplementary Data File 4? Supplementary Data File 3 reports elemental abundances. (`GCHRON-2023-15-Supp Data File-3-R80_02479-v01-Full Mass spec Proxigram_plot.xlsx`)

**306–9** This sentence is unclear, as Fig. 3 depicts model results. If this bears on the shape of high-to-low U transitions, it should be explained in more detail, or a different figure should be referenced.

**Figs. 1 & 2** Ideally present the same samples in each panel A and panel B rather than alternating.

**Fig. 3** I'm curious that it's a Normal distribution with $\sigma = 82$ nm that fits the profile. The standard deviation must be a function of the average R (40 nm), but is this mathematical relationship straightforward/quantifiable?

**Fig. 7** Please put panels A and B on equal x-axis scales to make comparing between the two panels easier.

**Fig. 9** Mention in the caption that the curve is a splined fit to help guide the eye and does not represent actual MSWD values.

**Fig. 11** Use consistent directionality in panels A & B (x increases in different directions).

---

## Referee Comment (RC2)

**Review of Denyszyn et al 2024: Short Communication: Nanoscale heterogeneity of U and Pb in baddeleyite – implications for nanogeochronology and 238U series alpha recoil effects**

**Summary**

This manuscript outlines atom probe tomography analysis of nanoscale variations in U and Pb distributions in two baddeleyite grains to assess the nuclear recoil distance from the decay of $^{238}$U to $^{206}$Pb. Results show that while one sample (Hart Dolerite) likely sampled a cleavage plane surface not a grain boundary so was not useful for assessing nuclear recoil. The other sample Great Dyke with some surface topography is likely a true grain boundary surface and shows a zonation of U, nanoscale clustering of U and a non-equilibrium $^{206}$Pb profile implying Pb distribution has been affected by nuclear recoil. Modelling of recoil distances making some reasonable assumptions of the array suggest that a recoil distance of 80-90 nm provides the best fit to the data, but this is larger than previous estimates of alpha recoil. The modelling partially overcomes the challenge with APT volumes being smaller in two/three dimensions than the alpha recoil distance. This is an important contribution to the field of geochronology and shows some fascinating new microstructures and features particularly at the nanoscale and their influence on isotope ratios used for radiometric dating using the U-Pb and Pb-Pb system.

Overall, the paper is generally clearly written, the APT data is excellent, and the discussion is open and honest clearly stating the limitations and assumptions and making testable predictions of how the authors ideas and inferences regarding anisotropic, microstructural and topographic effects on nuclear recoil may be evaluated in future work for which the authors should be commended. However, the figures and presentation of the data require improvement to bring them up to the same standard as the excellent, clear and detailed text descriptions.

As such I recommend this work for publication in Geochronology after minor tweaks to the text and major revisions to the figures.

**Figures**

Figure 1. This figure would benefit from labels and annotations of each feature of interest e.g. the baddeleyite crystals the Pt strip, sample mount. I would also propose removing the data bar below with the WD Det Curtin University etc., and instead remake the scale bar. The figure caption should state that it is the white rectangle as there are several rectangular features here.

Figure 2. As with figure one this figure would benefit from more annotations, e.g. baddeleyite, Si Post, Pt weld, and removing the data bar below with the WD Det Curtin University etc. and instead remake the scale bar.

Figure 3. The Y axis has no title to state what it is. This graph could also be tidied up by making the X-Y lines bold and removing the other grid lines. It is also a bit messy that the y data labels overlap the data. Distr in part B should be spelled out fully or the abbreviation defined in the caption and A and B panel names, and the graphic key should be on the same side of the graph between A and B.

Figure 4. See similar comments to figure 3. The y axis is undefined axis labels overlap data, x, y axis should be bold and grid lines removed to make the graph easier to access. The caption lacks detail and should be expanded.

Figure 5. Specimen M5 does not have a Cr cap. The implications of this should be discussed in the main text i.e. how sure can you be that you are indeed at a grain boundary. Or perhaps it does have a Cr cap that it is not visible. I propose adding a Cr ion map on to these datasets in a different colour to prove that you are indeed at the surface of the grain. ULF and HD in the figure need to be defined in the caption. The labels next to each dataset should be 238U and 206Pb not U and Pb. It is not clear from the caption where the crystal surface 0nm is and unless you are familiar with APT data this might not be clear. I propose you put a vertical line with distance markers that starts from 0 nm – grain surface and increases with depth into the sample. Further descriptions are required in the caption to describe the zoning of U and Pb in M5 and relative uniformity in M2.

Figure 6. As with previous figures the gridlines make the figure quite busy, I suggest removing them and making the X and y axis lines bolder. Again, the y axis requires a title to state what it is. Why does the X-axis title appear above the line here but below the axis in other plots. Remake so all graphs are consistent. I am glad to see that 2 sigma uncertainties were used in B however no note is made for the uncertainty in A, also a 2-sigma uncertainty this is inconsistent with the text which said 1 sigma was used. This should be checked and changed to consistently state what level of uncertainty was used. Either define measu extrapol in the caption or write it out in full. The captain lacks detail and should be expanded to describe fully what is being presented and any trends that can be observed.

Figure 7. faces the same issues as figure 6 and should be amended in the same way. In addition, it would be good to add the best fit lines and results of any statistical tests to part B to show that 80 nm is indeed the best fit to the measured ratio. Additionally, here R values of 40 80 and 120 are presented but in figure 3 20 nm, 30 nm and 40 nm R values were modelled and in figure 4 only a R value of 40 nm was modelled. I would include modelling results in figure 7 for 20, 30, and 40 nm here and include modelling results for 80 nm and 120 nm in Figure 3 and results for 20 nm, 30 nm 80nm and 120 nm in figure 4 for comparison. Additionally, the measured results do not have their associated error bars which should be included.

Figure 8. This figure is quite confusing. It faces similar issues to figures 6 and 7, which should be amended. In addition, the caption needs to be expanded substantially to explain what is being shown. i.e. that A is modelling the distribution of oscillatory zoning of U? and that part B is modelling $^{206}Pb/^{238}U$ for different R values assuming oscillatory zoning? Currently the caption only states that 40 nm R values were modelled in this way. Also, I would not describe this as a single high U zone? Please expand the caption to better explain the graphs. The caption should also state what the error bars represent.

Figure 9. this figure requires amendment as with figure 6 7 and 8. In addition it would be useful to plot the MSWD where no oscillatory zoning is assumed and instead the profile in Figure 7A is used for comparison of the goodness of the fit and the validity of the assumption of oscillatory zoning.

Figure 10. please provide a key for the isosurfaces in A and the dots in B also please add labels or annotations to the data to point out clusters of U atoms in both datasets. Expand the caption to explain what the data is showing.

Figure11. It is very confusing that panel A flips to have the 0 nm of the data to the right-hand side where all other data have the 0 nm or negative direction to the left hand side, please flip this figure so all data are in the same orientation. The key in A and the title of the graph do not match are you plotting isotopes or element concentrations. As well as the above please

amend this figure in a similar way to figure 6 and 7 with properly labelling axes and expanding captions to explain the data shown.

**Supplementary materials**

The supplementary materials require some additions.

To aid reproducibility the supplementary materials should present the APT operating conditions for all datasets in a table in line with the suggestions of Blum, T. B., Darling, J. R., Kelly, T. F., Larson, D. J., Moser, D. E., Perez-Huerta, A., ... & Valley, J. W. (2018). Best practices for reporting atom probe analysis of geological materials. Microstructural Geochronology: Planetary records down to atom scale, 369-373.

Not all data is presented in the supplementary materials consistently. For example, only sample M5 has a depth profile and mass spectrum while sample M2 has no data. Please provide an equivalent dataset for both samples.

I would suggest that the authors also provide the .RHIT files/raw data and range files so the analysis can be replicated.

**Minor comments:**

Line 16. For clarity, should state that the Hart Dolerite sample was likely a cleavage plane not a natural grain boundary.

Line 17. Delete 'apparently' as from the discussion there is almost nothing else this can be but a grain boundary or a fractured surface.

Line 114-120. The FIB preparation protocols used, and Cr capping are established sample preparation approaches and appropriate methods papers should be cited. E.g.:

 Thompson, K., Lawrence, D., Larson, D. J., Olson, J. D., Kelly, T. F., & Gorman, B. (2007). In situ site-specific specimen preparation for atom probe tomography. *Ultramicroscopy*, *107*(2-3), 131-139.

Daly, L., Lee, M. R., Hallis, L. J., Ishii, H. A., Bradley, J. P., Bland, P. A., ... & Thompson, M. S. (2021). Solar wind contributions to Earth's oceans. *Nature Astronomy*, *5*(12), 1275-1285.

Rickard, W. D., Reddy, S. M., Saxey, D. W., Fougerouse, D., Timms, N. E., Daly, L., ... & Jourdan, F. (2020). Novel applications of FIB-SEM-based ToF-SIMS in atom probe tomography workflows. *Microscopy and Microanalysis*, *26*(4), 750-757.

Line 136. Here and elsewhere, I believe it should be ionic species not specie. Specie refers to coins the singular of species is also species.

Line 137. Please state the charge state where no peaks were visible for $^{235}$U.

Line 140. Typically, it would be preferable to present isotope data with 2 sigma uncertainties rather than 1 to give confidence that the variation is natural and not due to analytical precision. Please also present the 2-sigma uncertainty in tables and figures.

Line 145. Please state the values of R that were assumed. Also, please rephrase so that it is clear that R is defined as the average value of each individual alpha recoil distance.

Line 176. What statistical test was used to evaluate the best fit of the modelled $^{206}$Pb/$^{238}$U curves vs the measured $^{206}$Pb/$^{238}$U curves?

Line 184. Please state the distance over which the U concentration varies from 150-650 ppma and state the uncertainties.

Line 185. The Cr cap is only apparently present in one APT dataset M2 not in M5. Please explain why this is the case and also how you can be sure the grain surface has been measured for M5 when no Cr capping layer is observed.

Line 186. Instead of 'these elements' please state specifically U and Pb.

Line 197. 'Very short' can you please quantify this.

Line 197. 'If not' is vague, please change to 'If the recoil distance was not very short, then'.

Line 230. 'to the right of the measured profile' is not specific as it doesn't give a reference frame from which to go right from. Please cite the figure referred to and direction of the high-U region relative to the x,y,z of the atom probe dataset.

Line 227. 'The best fit is 80 nm.' I agree by eye this appears to be true. However, could you present a statistical test of the closeness of the measured to modelled curves be presented to show that 80 nm is indeed the best candidate recoil distance. It appears later you do a MSWD for the oscillatory zoning can you present similar tests here?

Line 349. 'U concentration outside the surface was very low'. Out of interest would it be possible to calculate the minimum 'gap' between the grain surface/fracture surface and another U-bearing mineral that would produce the measured $^{206}Pb/^{238}U$ profile? Or how far away would another baddeleyite grain have to be to not impact the $^{206}Pb/^{238}U$ profile?

I hope these comments help the authors improve their manuscript ahead of publication.

Cheers

Luke

---

## Author Comment (AC1)

Author responses are in Red

Overview

Denyszyn and others couple atom probe tomography (APT) and numerical modeling to examine α-recoil processes in the U-Pb system of baddeleyite. They find that only one of the two samples (Great Dyke of Mauritania) shows any heterogeneity in U and Pb distribution, and they disregard the other sample (Hart dolerite), concluding that it is an interior region of a crystal exposed along a cleavage plane. Through APT, the authors identify both 238U and 206Pb profiles that reflect a combination of crystallization and recoil effects. They estimate a mean recoil length for the 238U-series (80–90 nm) that is larger than prior estimates and identify some plausible causes for this discrepancy.

Overall, I find the work and conclusions very sound, and I find this to be an important and relevant contribution to the U-Pb and U-series communities' understanding of recoil processes at a very fine scale 1. My primary criticism of this work is the lack of clarity and organization in the figures, captions, and some of the text, which collectively make it difficult to efficiently interpret the authors' findings. Because the scientific work is sound and the conclusions relevant to the field of geochronology, I would recommend the manuscript for publication in Geochronology if the following comments are sufficiently addressed. In addition to these specific comments, I encourage the authors to review the manuscript with special attention to clear figures, prose, and descriptive figure captions.

Graham H. Edwards

General Comments:
• The introduction is well-written and provides a very thorough background.
• I found it cumbersome to interpret figures with unlabelled y-axes. While the labels were floated in the plots as text boxes, the authors should label axes directly wherever possible (e.g. Figs. 3, 4, 6–9, 11) and position the y-axis and corresponding tick labels outside of the plot area to ensure they are as legible as possible (Figs. 3,4,7). I recognize that the authors prepared their figures in Excel, which offers limited customization capabilities, but all of these edits are possible in Excel and will significantly improve figure readability.

*All graphic figures have been redrafted to show labels and numbering outside the plot area as suggested by the reviewer.*

• The authors refer with some frequency to Supplementary Data, including three spreadsheets. While I take no issue with referring to this supplementary data, these spreadsheets are poorly curated and have inadequate metadata. In the case of Supplementary Data File 3, this does not appear to contain the data the authors describe on line 274. Moreover, Supplementary Data Files 2 and 4 are multi-page xlsm files. For clarity, the key plots should be presented as figures with descriptive captions

so that readers can efficiently interpret the authors' key points. I think it is good practice to include the xlsm files to illustrate the methodological process, but these should not be the primary format of presenting/describing nuanced supplementary data.
• For clarity in the results and discussion, I advise the authors to use a consistent and specific set of directions (e.g. edge, interior) that consistently describe orientation relative to the original sample rather than terms like "left" (lines 219–30) that are dependent on the orientation of the APT specimen.
• Some of the text-based figure captions (lines 510–41) do not correspond with the current figure numbers. In general, the authors should double check that captions and in-text references correspond with appropriate figures and supplementary data files.

*Supplementary Data File 3 does contain the results from modelling alpha recoil affected Pb/U profiles from U clusters. Explanatory sheets have been added to each of the Excel data files that should clarify their contents.*

• I find the use of a convolution of the U distribution with the redistribution distribution appropriate for estimating the distribution of recoil transported radiogenic Pb. However, I think there should be a more extended discussion at the beginning of §2.3 justifying this approach. Lines 168–9 give this a good start, but I think most readers would benefit from more detail on this specific method.

*The discussion in question (now at line 170) has been expanded to give a more detailed description of the convolution process.*

• The Cr caps appear to be at angles to the z-axis of the FIB-milled specimens. This is curious, as the authors model systems with the z-axis orthogonal to the crystal surface. The authors should comment on how the needle-shaped specimens are oriented relative to the surfaces of grains they were milled from. Are the caps just apparently skewed or does this reflect the angle of the needles relative to the crystal surface.

*The Cr cap of the Hart Dolerite sample (M2), is in this case at an angle. This is likely due to either irregularities in the surface topography of the sample at the nm scale, and/or the crystal surface not being exactly horizontal at the time of FIB milling. At any rate, the modelling was carried out on the Mauritania sample (M5) and not this specimen. A sentence to this effect has been added (line 120).*

• The authors conclude that the Hart dolerite is an exposed cleavage plane and do not consider it further. However, they consider spaces between bladed crystals as fast diffusion pathways of atom loss by recoil. They should comment on how these specific systems differ. (Presumably, the cleavage plane was exposed by a very recent breakage and was within strong crystal lattice previously, but I still think an explicit statement is worthwhile).

*(now at line 213)*
*We have now added "recently exposed cleavage plane" to clarify.*

Comments by line #:

97 I think this is based on the observations of Davis and Davis (2018), right? I think it's worth referencing the relevant study again with this statement.

*This reference has been added at line 109.*

201–10 Please mention how the abundances across the transect are calculated. Are they calculated from the entire disk of each depth bin? Also please mention that a distance of zero refers to the tip of the specimen (and the corresponding crystal surface). I could figure it out by comparing figs. 5 and 6A, but an explicit statement in the text and corresponding figure captions (e.g. 6) would be very helpful.

*This has been added at line 220. The discussion of measurements and modelling has now been split into two subsections (3.1 and 3.2).*

228–9 Please elaborate on the statement "but deviates from the measured profile near the start of the high U peak because of accumulation of recoiled Pb."

*This paragraph (now at line 245) has been revised at lines 260-261 and 264 to make the description clearer.*

230 Wouldn't 206Pb recoiled from the left be relatively inconsequential compared to Pb recoiled from the 238U peak? The process should be described in more detail.

*'Left' was mistakenly written instead of 'right'. The discussion has been corrected and clarified at lines 261-263.*

274 Supplementary Data File 4? Supplementary Data File 3 reports elemental abundances. (GCHRON-2023-15-Supp Data File-3-R80_02479-v01-Full Mass spec Proxigram_plot.xlsx)

*'Supplementary Data File 4' instead of 3 was mistakenly referred to at line 315 of the reviewed version. This has been corrected at new line 387. More description of the cluster data has also been given in lines 281-283 and 287.*

306–9 This sentence is unclear, as Fig. 3 depicts model results. If this bears on the shape of high-to-low U transitions, it should be explained in more detail, or a different figure should be referenced.

*Fig 3 should have been Fig 5. This has been corrected at new line 378.*

Figs. 1 & 2 Ideally present the same samples in each panel A and panel B rather than alternating.

*The figure labels are now consistent across the two figures.*

Fig. 3 I'm curious that it's a Normal distribution with σ = 82 nm that fits the profile. The standard deviation must be a function of the average R (40 nm), but is this mathematical relationship straightforward/quantifiable?

*Presumably it is but the mathematical reasoning behind the Central Limit Theorem is beyond the scope of the manuscript and probably the abilities of the authors, which is why Wikipedia is so useful. We have referenced Bárány and Vu, 2007 as the most recent publication on the subject.*

Fig. 7 Please put panels A and B on equal x-axis scales to make comparing between the two panels easier.

*In order to model 206Pb/238U over the range of measurement (panel A) it is necessary to use a U profile (panel B) that includes this range of measurement as well as extrapolated values above and below so that the calculation encompasses the total range of alpha recoil effects. The two are not meant to have the same scale, which is why they are split into separate panels. Panel B is extremely busy, as noted by a previous reviewer so we have used symbols that are as large and distinctive as possible. We feel that compressing it to the range of measured values in panel A would unnecessarily degrade its readability.*

Fig. 9 Mention in the caption that the curve is a splined fit to help guide the eye and does not represent actual MSWD values.

*This is now done in the caption.*

Fig. 11 Use consistent directionality in panels A & B (x increases in different directions).

*Fig 11 has been redrafted to accord with the reviewers suggestion, as well as to show other trace elements that are enriched in the clusters.*

---

## Author Comment (AC2)

Author response is in Red

Review of Denyszyn et al 2024: Short Communication: Nanoscale heterogeneity of U and Pb in baddeleyite – implications for nanogeochronology and 238U series alpha recoil effects

Summary

This manuscript outlines atom probe tomography analysis of nanoscale variations in U and Pb distributions in two baddeleyite grains to assess the nuclear recoil distance from the decay of $_{238}$U to $_{206}$Pb. Results show that while one sample (Hart Dolerite) likely sampled a cleavage plane surface not a grain boundary so was not useful for assessing nuclear recoil. The other sample Great Dyke with some surface topography is likely a true grain boundary surface and shows a zonation of U, nanoscale clustering of U and a non-equilibrium $_{206}$Pb profile implying Pb distribution has been affected by nuclear recoil. Modelling of recoil distances making some reasonable assumptions of the array suggest that a recoil distance of 80-90 nm provides the best fit to the data, but this is larger than previous estimates of alpha recoil. The modelling partially overcomes the challenge with APT volumes being smaller in two/three dimensions than the alpha recoil distance. This is an important contribution to the field of geochronology and shows some fascinating new microstructures and features particularly at the nanoscale and their influence on isotope ratios used for radiometric dating using the U-Pb and Pb-Pb system.

Overall, the paper is generally clearly written, the APT data is excellent, and the discussion is open and honest clearly stating the limitations and assumptions and making testable predictions of how the authors ideas and inferences regarding anisotropic, microstructural and topographic effects on nuclear recoil may be evaluated in future work for which the authors should be commended. However, the figures and presentation of the data require improvement to bring them up to the same standard as the excellent, clear and detailed text descriptions.

As such I recommend this work for publication in Geochronology after minor tweaks to the text and major revisions to the figures.

Figures

Figure 1. This figure would benefit from labels and annotations of each feature of interest e.g. the baddeleyite crystals the Pt strip, sample mount. I would also propose removing the data bar below with the WD Det Curtin University etc., and instead remake the scale bar. The figure caption should state that it is the white rectangle as there are several rectangular features here.

Thanks, the changes have been made

Figure 2. As with figure one this figure would benefit from more annotations, e.g. baddeleyite, Si Post, Pt weld, and removing the data bar below with the WD Det Curtin University etc. and instead remake the scale bar.

Thanks, the changes have been made

Figure 3. The Y axis has no title to state what it is. This graph could also be tidied up by making the X-Y lines bold and removing the other grid lines. It is also a bit messy that the y data labels overlap the data. Distr in part B should be spelled out fully or the abbreviation defined in the caption and A and B panel names, and the graphic key should be on the same side of the graph between A and B.

Fig 3 has been amended in accordance with both reviewer's comments.

Figure 4. See similar comments to figure 3. The y axis is undefined axis labels overlap data, x, y axis should be bold and grid lines removed to make the graph easier to access. The caption lacks detail and should be expanded.

Fig 4 has been amended in accordance with the reviewer's comments and the caption expanded to give more detail.

Figure 5. Specimen M5 does not have a Cr cap. The implications of this should be discussed in the main text i.e. how sure can you be that you are indeed at a grain boundary. Or perhaps it does have a Cr cap that it is not visible. I propose adding a Cr ion map on to these datasets in a different colour to prove that you are indeed at the surface of the grain. ULF and HD in the figure need to be defined in the caption. The labels next to each dataset should be 238U and 206Pb not U and Pb. It is not clear from the caption where the crystal surface 0nm is and unless you are familiar with APT data this might not be clear. I propose you put a vertical line with distance markers that starts from 0 nm – grain surface and increases with depth into the sample. Further descriptions are required in the caption to describe the zoning of U and Pb in M5 and relative uniformity in M2.

As seen in Figure 2, sample M5 did have a Cr cap. The caption mentioned that the measurement (starting at 0 nm) starts at the crystal surface, which was preserved under the cap. It's been re-written a bit to clarify. The scale bar is now more prominently displayed. The labelling issue was corrected as per Reviewer 1's comments. The caption has been expanded to refer the reader to the 3D video files in the Supplement for visualizing the distribution of U and Pb.

Figure 6. As with previous figures the gridlines make the figure quite busy, I suggest removing them and making the X and y axis lines bolder. Again, the y axis requires a title to state what it is. Why does the X-axis title appear above the line here but below the axis in other plots. Remake so all graphs are consistent. I am glad to see that 2 sigma uncertainties were used in B however no note is made for the uncertainty in A, also a 2-sigma uncertainty this is inconsistent with the text which said 1 sigma was used. This should be checked and changed to consistently state what level of uncertainty was used. Either define measu extrapol in the caption or write it out in full. The captain lacks detail and should be expanded to describe fully what is being presented and any trends that can be observed.

Fig 6 as well as 7 and 8 have been shown in the same way in accordance with the reviewer's comments. The caption has also been expanded to explain the significance of the profile. All error bars have been shown at 2 sigma.

Figure 7. faces the same issues as figure 6 and should be amended in the same way. In addition, it would be good to add the best fit lines and results of any statistical tests to part B to show that 80 nm is indeed the best fit to the measured ratio. Additionally, here R values of 40 80 and 120 are presented but in figure 3 20 nm, 30 nm and 40 nm R values were modelled and in figure 4 only a R value of 40 nm was modelled. I would include modelling results in figure 7 for 20, 30, and 40 nm here and include modelling

results for 80 nm and 120 nm in Figure 3 and results for 20 nm, 30 nm 80nm and 120 nm in figure 4 for comparison. Additionally, the measured results do not have their associated error bars which should be included.

Error bars have been added to the data on Fig 7 and the format amended. The caption has been expanded to explain the graph in more detail.

Figure 8. This figure is quite confusing. It faces similar issues to figures 6 and 7, which should be amended. In addition, the caption needs to be expanded substantially to explain what is being shown. i.e. that A is modelling the distribution of oscillatory zoning of U? and that part B is modelling $^{206}Pb/^{238}U$ for different R values assuming oscillatory zoning? Currently the caption only states that 40 nm R values were modelled in this way. Also, I would not describe this as a single high U zone? Please expand the caption to better explain the graphs. The caption should also state what the error bars represent.

The format of Fig 8 has been amended. The caption has been expanded to explain the graph in more detail.

Figure 9. this figure requires amendment as with figure 6 7 and 8. In addition it would be useful to plot the MSWD where no oscillatory zoning is assumed and instead the profile in Figure 7A is used for comparison of the goodness of the fit and the validity of the assumption of oscillatory zoning.

The format of Fig 9 has been amended and the results for both oscillatory zoning and a single zone are shown.

Figure 10. please provide a key for the isosurfaces in A and the dots in B also please add labels or annotations to the data to point out clusters of U atoms in both datasets. Expand the caption to explain what the data is showing.

We have modified the figure caption to accommodate the reviewer's request. The caption now reads: Figure 10: (A) Distribution of 0.2 at% U isoconcentration surfaces (approximately ten times background) delineating U clusters in sample M5. (B) Each yellow sphere represents a single U atom in a 40 x 40 x 20 nm sub-domain, illustrating the clustered distribution of U.

Figure11. It is very confusing that panel A flips to have the 0 nm of the data to the right-hand side where all other data have the 0 nm or negative direction to the left hand side, please flip this figure so all data are in the same orientation. The key in A and the title of the graph do not match are you plotting isotopes or element concentrations. As well as the above please amend this figure in a similar way to figure 6 and 7 with properly labelling axes and expanding captions to explain the data shown.

Fig 11 has been amended so both sub-figures have the X-axis in the same direction. Titles and caption have been revised to be clearer and the format amended as with the other graphs.

Supplementary materials

The supplementary materials require some additions.

To aid reproducibility the supplementary materials should present the APT operating conditions for all datasets in a table in line with the suggestions of Blum, T. B., Darling, J. R., Kelly, T. F., Larson, D. J., Moser, D. E., Perez-Huerta, A., ... & Valley, J. W. (2018). Best practices for reporting atom probe analysis of geological materials. Microstructural Geochronology: Planetary records down to atom scale, 369-373.

Not all data is presented in the supplementary materials consistently. For example, only sample M5 has a depth profile and mass spectrum while sample M2 has no data. Please provide an equivalent dataset for both samples.

I would suggest that the authors also provide the .RHIT files/raw data and range files so the analysis can be replicated.

We have included a new supplementary data file reporting the APT analytical and reconstruction details for both datasets. The new file is named Supplementary File 7. However, we have not included a new mass spectrum. The spectra currently reported in supplementary materials is representative of both datasets. Adding yet another figure would arguably confuse the reader and doesn't add anything to the data evaluation. In response to the request to share the raw data, we respectfully decline this request – it would be of limited use to most readers and bei n the form of 1.8 Gb files. Perhaps best considered "on request".

Minor comments:

Line 16. For clarity, should state that the Hart Dolerite sample was likely a cleavage plane not a natural grain boundary.

Thanks, done

Line 17. Delete 'apparently' as from the discussion there is almost nothing else this can be but a grain boundary or a fractured surface.

Thanks, done

Line 114-120. The FIB preparation protocols used, and Cr capping are established sample preparation approaches and appropriate methods papers should be cited. E.g.:

Thompson, K., Lawrence, D., Larson, D. J., Olson, J. D., Kelly, T. F., & Gorman, B. (2007). In situ site-specific specimen preparation for atom probe tomography. *Ultramicroscopy*, *107*(2-3), 131-139.

Daly, L., Lee, M. R., Hallis, L. J., Ishii, H. A., Bradley, J. P., Bland, P. A., ... & Thompson, M. S. (2021). Solar wind contributions to Earth's oceans. *Nature Astronomy*, *5*(12), 1275-1285.

Rickard, W. D., Reddy, S. M., Saxey, D. W., Fougerouse, D., Timms, N. E., Daly, L., ... & Jourdan, F. (2020). Novel applications of FIB-SEM-based ToF-SIMS in atom probe tomography workflows. *Microscopy and Microanalysis*, *26*(4), 750-757.

Thanks, we have added the references.

Line 136. Here and elsewhere, I believe it should be ionic species not specie. Specie refers to coins the singular of species is also species.

Agreed, done

Line 137. Please state the charge state where no peaks were visible for $_{235}$U.

It now clarifies that "No peaks were visible above background for any species of $^{235}$U."

Line 140. Typically, it would be preferable to present isotope data with 2 sigma uncertainties rather than 1 to give confidence that the variation is natural and not due to analytical precision. Please also present the 2-sigma uncertainty in tables and figures.

We have shown 2 sigma error bars on all figures.

Line 145. Please state the values of R that were assumed. Also, please rephrase so that it is clear that R is defined as the average value of each individual alpha recoil distance.

New line 255. This has been done.

Line 176. What statistical test was used to evaluate the best fit of the modelled 206Pb/238U curves vs the measured 206Pb/238U curves?

It is explained at line 256 and below that the 80 nm distance gives a better visual fit than 40 nm and 120 nm and that this is confirmed by the MSWD values, which are now plotted on Fig 9 for both oscillatory U zoning and a single zone.

Line 184. Please state the distance over which the U concentration varies from 150-650 ppma and state the uncertainties.

Thanks, done

Line 185. The Cr cap is only apparently present in one APT dataset M2 not in M5. Please explain why this is the case and also how you can be sure the grain surface has been measured for M5 when no Cr capping layer is observed.

This refers to Fig 2, which does show the Cr cap present for both specimens.

Line 186. Instead of 'these elements' please state specifically U and Pb.

Done as per Reviewer 1

Line 197. 'Very short' can you please quantify this.

Done to "<5 nm", consistent with the statement in the first paragraph of the following section.

Line 197. 'If not' is vague, please change to 'If the recoil distance was not very short, then'.

Thanks, done

Line 230. 'to the right of the measured profile' is not specific as it doesn't give a reference frame from which to go right from. Please cite the figure referred to and direction of the high-U region relative to the x,y,z of the atom probe dataset.

Clarified to "deeper within the crystal", and refers to Fig. 8A in the next sentence.

Line 227. 'The best fit is 80 nm.' I agree by eye this appears to be true. However, could you present a statistical test of the closeness of the measured to modelled curves be presented to show that 80 nm is indeed the best candidate recoil distance. It appears later you do a MSWD for the oscillatory zoning can you present similar tests here?

New Line 256. Please see reply to comment on line 176 above.

Line 349. 'U concentration outside the surface was very low'. Out of interest would it be possible to calculate the minimum 'gap' between the grain surface/fracture surface and another U-bearing mineral that would produce the measured $_{206}Pb/_{238}U$ profile? Or how far away would another baddeleyite grain have to be to not impact the $_{206}Pb/_{238}U$ profile?

New line 389. This would require that the distance between grains be much less than the average recoil distance. We think that this comment bears on the question of whether the Hart dolerite grain, which showed no decrease in Pb concentration near the edge, may have been situated next to another baddeleyite grain. We have not discussed this because it is extremely unlikely for a random distribution of baddeleyite crystals and it would require that the adjacent crystal have a similar U concentration.

I hope these comments help the authors improve their manuscript ahead of publication.

Cheers

Luke